# Myocardin-Related Transcription Factor Mediates Epithelial Fibrogenesis in Polycystic Kidney Disease

**DOI:** 10.3390/cells13110984

**Published:** 2024-06-05

**Authors:** Zsuzsanna Lichner, Mei Ding, Tarang Khare, Qinghong Dan, Raquel Benitez, Mercédesz Praszner, Xuewen Song, Rola Saleeb, Boris Hinz, York Pei, Katalin Szászi, András Kapus

**Affiliations:** 1Keenan Research Centre for Biomedical Science, St. Michael’s Hospital, Toronto, ON M5B 1T8, Canada; lichner_zsuzsi@yahoo.com (Z.L.); tarang.khare@gmail.com (T.K.); rola.saleeb@unityhealth.to (R.S.); katalin.szaszi@unityhealth.to (K.S.); 2Enrich Bioscience, Toronto, ON M5B 1T8, Canada; 3Division of Nephrology, University Health Network, Toronto, ON M5G 2C4, Canada; 4Department of Laboratory Medicine and Pathobiology, Temerty School of Medicine, University of Toronto, Toronto, ON M5S 1A8, Canada; 5Faculty of Dentistry, University of Toronto, Toronto, ON M5G 1G6, Canada; 6Department of Surgery, University of Toronto, Toronto, ON M5T 1P5, Canada; 7Department of Biochemistry, University of Toronto, Toronto, ON M5S 1A8, Canada

**Keywords:** ∙ polycystic kidney disease, epithelium-initiated fibrosis, myocardin-related transcription factor, profibrotic epithelial phenotype

## Abstract

Polycystic kidney disease (PKD) is characterized by extensive cyst formation and progressive fibrosis. However, the molecular mechanisms whereby the loss/loss-of-function of Polycystin 1 or 2 (PC1/2) provokes fibrosis are largely unknown. The small GTPase RhoA has been recently implicated in *cystogenesis*, and we identified the RhoA/cytoskeleton/myocardin-related transcription factor (MRTF) pathway as an emerging mediator of epithelium-induced fibrogenesis. Therefore, we hypothesized that MRTF is activated by PC1/2 loss and plays a critical role in the fibrogenic reprogramming of the epithelium. The loss of PC1 or PC2, induced by siRNA in vitro, activated RhoA and caused cytoskeletal remodeling and robust nuclear MRTF translocation and overexpression. These phenomena were also manifested in PKD1 (RC/RC) and PKD2 (WS25/−) mice, with MRTF translocation and overexpression occurring predominantly in dilated tubules and the cyst-lining epithelium, respectively. In epithelial cells, a large cohort of PC1/PC2 downregulation-induced genes was MRTF-dependent, including cytoskeletal, integrin-related, and matricellular/fibrogenic proteins. Epithelial MRTF was necessary for the paracrine priming of the fibroblast–myofibroblast transition. Thus, MRTF acts as a prime inducer of epithelial fibrogenesis in PKD. We propose that RhoA is a common upstream inducer of both histological hallmarks of PKD: cystogenesis and fibrosis.

## 1. Introduction

Autosomal dominant polycystic kidney disease (hereafter PKD) is the most common renal genetic disorder, with a prevalence of 1:500–1000 [1,2]. PKD is caused by loss-of-function mutations in Polycystin 1 (PC1, encoded by *PKD1*, 85%) or Polycystin 2 (PC2, encoded by *PKD2*, 15%). They can form a heteromultimer and act as a chemo- and mechanosensitive complex [3,4], impacting a variety of functions, including cell division and metabolism [5,6,7]. Histologically, the major features of PKD are fluid-filled cysts and progressive renal fibrosis, which ultimately ruin the kidney architecture, leading to End-Stage Renal Disease (ESRD) [8,9]. Intensive work has uncovered the fundamental mechanisms in cystogenesis, linking PC1/PC2 loss-of-function to multiple signaling pathways [10,11,12,13,14]. However, much less is known about the mechanism underlying the accompanying fibrogenesis.

PKD-related fibrosis is a pathologically important biological puzzle: the disease is epithelium-initiated, but affects the mesenchymal compartment. Recent work, including our own, has shown that epithelial injury is a key causative factor in renal fibrosis [15,16,17,18,19,20]. Injured epithelial cells undergo partial epithelial–mesenchymal transition, acquiring a profibrotic epithelial phenotype (PEP) [15]. This state is characterized by cytoskeletal remodeling and the production of profibrotic cytokines/matricellular proteins, which, in turn, induce the surrounding mesenchyme to become matrix-producing fibroblasts and contractile myofibroblasts [15,21,22,23,24]. We and others have shown that MRTF-A and its isoform MRTF-B (MRTF) are key mediators of the initial epithelial reprogramming and consequent mesenchymal response [15,25,26,27,28,29,30,31,32] (reviewed in [33]). Under resting conditions, MRTF resides mostly in the cytosol because it binds G-actin, which masks its nuclear localization sequence (NLS) and promotes its nuclear efflux [34,35,36]. Cellular injury triggers actin polymerization, primarily mediated by the activation of Rho GTPases, such as RhoA. This leads to the dissociation of G-actin from MRTF and its consequent nuclear translocation. In the nucleus, MRTF binds its cognate transcription factor partner, serum response factor (SRF), and the complex drives a set of genes encoding cytoskeletal, matricellular/matrix proteins, and soluble fibrogenic mediators [15,25,27,28,37,38,39,40,41]. In addition, SRF can be switched on by an MRTF-independent mechanism via the ternary complex factors (TCFs) [42], leading to the activation of immediate early genes (c-Fos and c-Myc) regulating cell cycle entry and proliferation, relevant for PKD pathogenesis [43]. 

Remarkably, enhanced RhoA signaling has been recently implicated in PKD as an important input for *cystogenesis*. PC1 loss-of-function was shown to induce RhoA activation [44,45], which, in turn, triggered the Rho kinase- and myosin phosphorylation-dependent activation of the Hippo pathway effector YAP and its paralog TAZ, facilitating cyst formation. c-Myc was identified as a critical YAP target gene driving *cystogenesis* [44]. This scenario raised the intriguing possibility that MRTF, downstream of activated RhoA signaling, might mediate the other major arm: epithelial fibrogenesis. In further support of this hypothesis, a subset of MRTF and YAP target genes is common [15,30,46,47,48], both MRTF and YAP/TAZ contribute to PEP [15,29], and MRTF itself is an upstream transcription factor for TAZ [21,49]. Further, a systems biology approach identified SRF as a major dysregulated transcription factor pathway in PKD [50]. However, it remains to be elucidated whether MRTF is activated/translocated upon PC1/2 loss or in PKD, and whether it is a significant contributor to a fibrogenic signature. Accordingly, our goal was to place MRTF on the “PKD map” and to link it to fibrogenesis. 

Our results show that absent/dysregulated PC1 and PC2 signaling induces the nuclear translocation of MRTF in tubular epithelial cells in a RhoA-dependent manner. MRTF, in turn, facilitates the epithelial expression of cytoskeletal components, integrins, matricellular proteins, and other mediators, thereby priming mesenchymal cells for fibrogenesis. 

## 2. Materials and Methods

### 2.1. Cell Lines and Reagents

LLC-PK1 (clone 4), a porcine proximal tubule cell line, was a gift from Peter Harris (Vanderbilt University School of Medicine, TN) and was used as in our previous studies [28]. Briefly, the cells were cultured in DMEM (11885084, Gibco, Waltham, MA, USA), supplemented with 10% FBS and penicillin–streptomycin (Life Technologies, Carlsbad, CA, USA). Transforming growth factor β1 ((TGFβ), R&D, Minneapolis, MN, USA) treatment was carried out for 24 h in serum-free DMEM media at a 5 ng/mL concentration. 

### 2.2. siRNA-Mediated Silencing

siRNA transfections were performed using Lipofectamine RNAiMax (ThermoFisher Scientific, Waltham, MA, USA), following the manufacturer’s protocol. The cells were transfected at 30–40% confluency. Twenty-four hours post-transfection, the cells were serum-starved for an additional 24 h, and were assayed at 48 h post-transfection. siRNAs were used at the 100 nM final concentration, with the exception of siPC1 (150 nM), siRhoA (50 nM), and siMRTF-A (50 nM). The non-related control siRNA (siNR, Silencing Select Negative Control 1, ThermoFisher Scientific) concentration was matched in each experiment.

### 2.3. Real-Time Quantitative PCR (RT-qPCR)

The total RNA was isolated with RNeasy Mini Kit (Qiagen, Venlo, The Netherlands), following the manufacturer’s protocol. In total, 500–1000 ng of RNA was reverse-transcribed using the iScript Reverse Transcription Supermix (Bio-Rad, Montreal, QC, Canada), and gene expression was quantified using the PowerUP Syber Master Mix (Thermo Fisher Scientific, Waltham, MA, USA) on the QuantStudio 7 Pro system (ThermoFisher Scientific). Relative gene expression was calculated by the ∆∆ Ct method. A melt curve analysis confirmed the amplification of a single product, Table 1.

### 2.4. Western Blot Analysis

Western blot analysis was performed, as described before [28,51], on nitrocellulose membrane. The blots were visualized by Clarity or Clarity Max ECL Western Blotting Substrate (Bio-Rad Laboratories) and Chemidoc Imaging Systems (Bio-Rad Laboratories). Relative protein expression was calculated by ImageLabs (Bio-Rad). Horseradish peroxidase-conjugated secondary antibodies (1:2000) were from Jackson ImmunoResearch Laboratories. The following primary antibodies were used: Polycystin 1 (1:200 sc-130554, Santa Cruz Biotechnology, Dallas, TX, USA), Polycystin 2 (1:500, sc-28331, Santa Cruz Biotechnology), MRTF-A (1:1000, s14760 Cell Signaling Technology, Danvers, MA, USA), and total RhoA (1:1000, 2117S Cell Signaling Technology). GAPDH (sc-47724, Santa Cruz Biotechnology and 6004-1-Ig, Proteintech, Rosemont, IL, USA) or β-actin (A1978, Sigma Aldrich, Burlington, MA, USA) antibodies were used for normalization.

### 2.5. Preparation of GST-Fusion Protein and Rho Activation Assay

The preparation of GST-RBD (RhoA-binding domain (RBD): amino acids 7–89 of Rhotekin) and the Rho affinity assay were described in [52]. The beads were stored frozen in the presence of glycerol. Confluent LLC-PK1 cells were transfected with PC1 or 2 siRNA, and 48 h later, lysed with ice-cold assay buffer containing 100 mM NaCl, 50 mM Tris base (pH 7.6), 20 mM NaF, 10 mM MgCl_2_, 1% Triton X-100, 0.5% deoxycholic acid, 0.1% SDS, 1 mM Na_3_VO_4_, and protease inhibitors. The lysates were centrifuged, and aliquots for determining the total RhoA were removed. The remaining supernatants were incubated with 20–25 µg of GST-RBD at 4 °C for 45 min, followed by extensive washing. Aliquots of total cell lysates and precipitated proteins were analyzed by Western blotting and quantified by densitometry. Precipitated (active) RhoA was normalized using the corresponding total cell lysates.

### 2.6. Immunofluorescence Microscopy and Quantification

Cells were grown on glass coverslips for visual expression analysis or on 96-well plates (Corning) for quantitative image analysis with the ImageXpress Micro 4 System (Molecular Devices, San Jose, CA, USA). The cells were transfected as detailed above. Immunofluorescence staining was performed as described [53]. The following primary antibodies were used: MRTF-A (1:300), active RhoA (1:100, New East Biosciences, King of Prussia, PA, USA), and active ITGB1 12G10 (1:100 ab150002, Abcam, Cambridge, UK). F-actin was visualized by staining with fluorescently labelled phallodin (Phalloidin iFLUOR 555, Abcam) at 1:10,000 dilution for 1 h. The cells were imaged by either a WaveFX spinning-disk microscopy system (Quorum Technologies, Eugene, OR, USA) equipped with an ORCA-Flash4.0 digital camera or by an Olympus IX81 microscope with the Evolution QEi Monochrome camera, both driven by the MetaMorph 7.8 software (Molecular Devices, San Jose, CA, USA). Nuclear translocation was assessed by ImageXpress, driven by MetaExpress software’s inbuilt Multi Wavelength Translocation analysis module, as in our previous work [53]. Briefly, nuclear staining was measured as the mean fluorescence intensity of MRTF-A-specific staining within the DAPI-positive nuclei, and cytoplasmic staining was measured as the MRTF-A-positive fluorescence intensity in a preset ring around the nuclei. The nuclear/cystoplasmic ratio of MRTF-A was arranged in bins incremented by 0.02 (x axis). Active integrin clusters were counted by the MetaMorph software using the Manually Count Objects option. Counts were normalized to the cell number.

### 2.7. Next-Generation Sequencing Transcriptome Analysis

mRNA libraries were prepared using the NEBNext^®^ Poly(A) mRNA Magnetic Isolation Module, NEBNext^®^ Ultra™ II Directional RNA Library Prep with Sample Purification Beads and NEBNext Multiplex Oligos for Illumina (96) (New England Biolabs, Ipswich, MA, USA). Sequencing was carried out on the NovaSeq SP flowcell SR200 or PE100 (Illumina, San Diego, CA, USA). The data analysis is detailed in Appendix B.

### 2.8. LLC-PK1 and Fibroblast Communication, Collagen Substrate Wrinkling Quantification

Wild-type subcutaneous fibroblasts (WT SCF) were isolated from C57BL/6 mice and cultured on soft substrates with a Young’s elastic modulus of 0.2 kPa. The substrates were generated as described. Subsequently, the substrates were oxygenized and coated with gelatin (2 µg/cm^2^ diluted in PBS) [54]. In total, 2.000 SCF/cm^2^ were seeded and maintained in Dulbecco’s modified Eagle’s medium (DMEM) supplemented with 10% fetal bovine serum (FBS) and 1% penicillin–streptomycin for 6 h prior to synchronization with serum-free DMEM overnight. Conditioned media (CM) was obtained from LLC-PK1 PK1 renal epithelial cells. The epithelial cells were transfected with the indicated siRNAs. Twenty-four hours post-transfection, the media was removed, the cells were washed three times, and fresh serum-free media was added. After 24 h (48 h post-transfection), the CM was collected. Fibroblasts were stimulated with the CM for 48 h. Subsequently, phase-contrast images were taken of the fibroblasts by using a Zeiss Axio Observer Microscope. Cell contractile function, related to % area covered by wrinkles, was analyzed by ImageJ-win64 software (Madison, WI, USA) and the values were normalized to the cell number. 

### 2.9. Animal Tissues and Patient Specimens

Animals were bred and housed at the Toronto General Hospital Research Institute, University Health Network (UHN). Paraffin-embedded blocks of the kidneys of 12-month-old PKD1 RC/RC (n = 3) and control littermates (WT, n = 3), as well as 3-month-old PKD2 WS25/− (n = 6) and control littermates (PKD2 WS25/+, n = 3) [44,55,56], were processed at the Keenan Research Centre for Biomedical Sciences, St Michael’s Hospital. 

Nephrectomy specimens were collected at St Michael’s Hospital and included samples from PKD and RCC patients, following their informed consent. 

### 2.10. Immunohistochemistry and Quantification

Paraffin-embedded sections were stained as described. Antigen retrieval was performed by boiling the section for 5 min in TE buffer. The staining was viewed by an Olympus BX50 microscope, using the Cellsense 1.14 software (Olympus LS, Tokyo, Japan). Stained sections were scanned with Axio Scan Z1 (Zeiss, Jena, Germany) driven by the integrated ZEN Slidescan software, and analyzed by HALO v2.3.2089.23, using the inbuilt Area Quantification and Object Quantification modules. For quantifying CTGF and PDGF-B, 3 animals in each group were used, and DAB staining was quantified by HALO in 5–10 large randomly selected fields. DAB OD was normalized to the number of nuclei in each area. For quantitation of MRTF-A staining, thresholds were set to recognize hematoxylin stain (nuclei) and DAB (MRTF-A), and the average DAB staining intensity was determined for each nucleus. The total scale of DAB OD was divided into 256 evenly distributed intensity bins (X axis), and the frequency for each intensity bin was calculated in Excel. Six PKD2 WS25/− animals and six controls (PKD2 WS25/+ littermates) were analyzed, however one PKD2 WS25/− animal was removed from the quantification because it did not develop renal cysts. Tubules with a diameter over 1.5× the diameter of normal tubules were classified as dilated. Microcysts had flattened epithelial lining and, in general, lacked brush border.

### 2.11. RNAScope

Three-month-old PKD2 WS25/− (n = 3) and control PKD2 WS25/+ littermates (n = 3) were analyzed for MRTF-A and CTGF mRNA expression using 3 μm thin sections of paraffin-embedded kidneys. Staining was carried out by closely following the manufacturer’s protocol; 10% Gill’s hematoxylin was used to stain the nuclei. The stained sections were scanned by Axio Scan Z1 and analyzed using the HALO software v2.3.2089.23. Renal tubules with normal and dilated morphology (n = 15) and cysts (n = 5) were selected for analysis. Thresholds were set for hematoxylin, MRTF-A, and CTGF staining. OD was normalized to the number of nuclei.

### 2.12. Statistical Analysis

Immunofluorescence and Western blot images show representative results of a minimum of three similar experiments or the number of experiments indicated (n). Graphs show the means ± standard error of the mean. Statistical significance was determined by Student’s *t*-test or one-way analysis of variance (Tukey post hoc testing), using Excel2021 or Prism v7.0 softwares (Microsoft, Redmond, WA, USA). Violin plots were generated with the statskingdom.com/violin-plot-maker.html website, including the median and excluding outliers (Tukey) options. Statistical significance was calculated using a one-sample or two-tailed *t*-test, as appropriate. *p* < 0.05 was accepted as significant. Unless indicated otherwise, *, **, and *** correspond to *p* < 0.05, <0.01, and <0.001, respectively.

## 3. Results

### 3.1. PC1 or PC2 Downregulation Activates RhoA and Induces Nuclear Translocation of MRTF

To test whether PC1 or PC2 downregulation affected MRTF signaling, we silenced the corresponding genes in LLC-PK1 tubular cells, using specific siRNAs. This approach efficiently and selectively reduced the expression of the corresponding proteins, allowing for their independent manipulation (Figure 1A and Appendix A). 

To assess whether these proteins impact RhoA activity in our cellular system, we measured the level of active (GTP-bound) RhoA using a GST-Rhotekin pull-down assay, as previously performed [52]. Active RhoA (normalized to total RhoA) was significantly elevated upon PC1 loss [44,45], and similar results were obtained for PC2 (Figure 1B). Quantitative immunofluorescence staining with an active RhoA-specific antibody corroborated these results (Figure 1C). Loss of PC1 or PC2 resulted in dramatic cytoskeletal remodeling, characterized by the formation of strong actin stress fibers (Figure 1D). This was accompanied by a substantial nuclear translocation of MRTF. Both changes were prevented/mitigated by the concomitant downregulation of RhoA (Figure 1D). These qualitative observations were quantified by two means. First, by visual inspection, using a tripartite compartmental distribution (cytosolic, nuclear, or both/even) of MRTF. While MRTF was predominantly cytosolic under control conditions, it exhibited a significant shift to the nucleus upon the loss of PC1 or PC2. This was reversed by RhoA downregulation (Figure 1E). Second, to overcome regional heterogeneity, the distribution of single-cell nuclear/cytosolic MRTF ratios was automatically determined in large cell populations (detailed in Section 2). PC1 or PC2 silencing resulted in a shift toward higher N/C ratios (black vs. red curves), which was strongly mitigated by RhoA silencing (Figure 1F). The cumulative frequency of N/C ratios ≥1.4 (univocally assessed as “nuclear MRTF” by visual inspection) showed a ≈6-fold rise in PC1- or PC2-silenced cells relative to controls; this change was abolished by RhoA downregulation (Figure 1G). Together, these results show that the loss of PC1 or PC2 induces robust RhoA-dependent MRTF translocation into the nucleus. 

### 3.2. PC1/2 Loss Elevates MRTF Expression 

During these experiments, we noted that PC loss affected not only the distribution, but also the total expression of MRTF. Indeed, MRTF-A immunostaining significantly increased both in PC1- and PC2-silenced cells (Figure 2A,B), a finding confirmed by the Western blots (Figure 2C). This rise was, at least in part, due to increased MRTF gene transcription, since PC1 and PC2 downregulation significantly elevated the message for both *MRTF-A* and *MRTF-B* (Figure 2D). These findings imply that PC1/2 loss facilitates MRTF signaling both at the level of activation/nuclear translocation and that of total expression. 

### 3.3. MRTF in Action upon PC Loss in the Epithelium: A Targeted Approach 

Next, we addressed the functional significance of the observed changes. As an initial strategy, we concentrated on genes that satisfied each of the following criteria: they were shown to be (a) PC-sensitive [50,57], (b) MRTF targets [37], and (c) involved in fibrogenesis/PEP. The downregulation of PC1 (Figure 3A, left panels) or PC2 (Figure 3A, right panels) significantly enhanced the mRNA expression for matricellular proteins/fibrogenic mediators, such as transgelin (TAGLN) (see also Appendix A for alternative PC1 and PC2 siRNAs), CTGF, and CYR1. The downregulation of MRTF-A strongly reduced *TAGLN* and *CTGF* expression upon *PC1* or *PC2* silencing, and significantly decreased *CYR61* expression in *PC1*- but not in *PC2*-downregulated cells (Figure 3A). Efficient *MRTF-A* downregulation was not altered by the concomitant silencing of *PC1* or *PC2* (Appendix A). The impact of MRTF-A was preserved when *PC1/2*-downregulated cells were exposed to TGFβ1, the most potent fibrogenic cytokine. Of note, TGFβ1 is elevated in PKD and plays a pathogenic role in the disease [58,59]. 

TGFβ1 potentiated/augmented the effect of PC1/PC2 loss for each of these three genes, and MRTF significantly reduced the combined effect of these stimuli, except for PC2-downregulation-induced *CYR61* expression (Figure 3A). PC1 or PC2 loss also stimulated the expression of *ANXA1* and *RASSF2* in an MRTF-A-dependent manner (Figure 3B), although these were not stimulated by TGFβ1. We used the most responsive gene, *TAGLN*, to test RhoA and SRF dependence as well. The downregulation of either efficiently reduced PC1/2 loss-provoked *TAGLN* expression (Figure 3C,D). MRTF-B also contributed to these responses, because its silencing also mitigated the PC1- or PC2-loss-triggered increase in *TAGLN*, *CTGF*, and *CYR61*. Interestingly, in *PC2*-downegulated cells, *CYR61* mRNA expression was selectively sensitive to MTRF-B depletion (Figure 3E,F). 

Next, we quantified mRNA expression for some pro-inflammatory genes, aware that MRTF can physically associate with and inhibit NFκβ [60,61]. PC2 downregulation stimulated *TNFα, CCL2*, and *IL1β* expression, while it did not alter *IL1α* mRNA levels. MRTF-A silencing significantly *potentiated* the effect of PC2 loss on TNFα and IL1β and increased *IL1α* mRNA expression (Figure 3G). Thus, MRTF is an important contributor to the PC-loss-induced expression of key fibrogenic/PEP genes (Figure 3F) and a suppressor of some proinflammatory genes, shifting the balance from acute epithelial inflammatory responses to fibrosis. 

### 3.4. MRTF in Action upon PC Loss: A General Approach

To assess the role of MRTF in the molecular pathogenesis of PKD from a wider angle, we performed RNA-Seq. We focused on genes that (a) were upregulated upon PC1 *or* PC2 silencing *and* (b) were MRTF-dependent. Three complementary analysis methods were used to identify such PKD-related and MRTF-dependent events (Figure 4A and Appendix B). First, we identified differentially expressed genes/transcripts (DEG) that were significantly elevated upon *PC1* or *PC2* downregulation *and* significantly suppressed by concomitant *MRTF* silencing. The corresponding 130 (PC1/MRTF) and 128 (PC2/MRTF) transcripts are shown in (Figure 4B). Common enriched biological processes included epithelial cell migration, wound healing, and cell contractility, in line with the key role of MRTF in early epithelial injury responses (Figure 4C and Appendix A).

Second, Gene Set Enrichment Analysis (GSEA) (Figure 4D) indicated that the actomyosin cytoskeleton was among the most significant MRTF-dependent PKD-associated GO terms. The presence of closely related categories (microtubules and supramolecular polymers) underlines the relationship between PKD, cytoskeletal reorganization, and MRTF. GSEA also identified the ‘mitochondrial matrix’ and ‘organelle assembly’ as MRTF-supported categories. The MRTF dependence of these is of special interest, as altered cellular metabolism is a hallmark of PKD [62,63]. Moreover, MRTF emerged as a significant negative regulator of the early immune response.

Third, we utilized Weighted Gene Co-expression Network Analysis (WGCNA) to identify unbiased positive and negative correlation patterns. Among the resulting 90 modules, 6 matched our expression criteria, supplemented with the extra-requirement that PC1 *and* PC2 loss should act concordantly. 

Of these, ME18 was ranked first, and it was the second most significant among all the 90 modules (Figure 4E). ME18-related biological processes reflected the previously identified themes (cell junction, microtubules, cytoskeletal organization, cell projection assembly, and cell motility) (Figure 4F).

Considering the common cis-elements in the target genes, SRF-binding sites were highly enriched in the MRTF-dependent, PKD-related gene promoters, as expected. Interestingly, the recognition elements of innate-immune-response-related transcription factors (particularly NFκB) were also significantly enriched, concordant with the DEG analysis, suggesting that MRTF inhibits NFκB (Figure 4G–H).

Finally, we confirmed the PC1/2 status-independent downregulation of *MRTF-A* by siRNA and the concomitant suppression of the PC1/2 loss-promoted *TAGLN* and PC1-specific response of *CYR61* [64,65,66]. In addition, we also showed the MRTF-dependent behavior of *Col4A2*, a basement membrane component, the expression of which was shown to increase in renal fibrosis [15] (Figure 4H). Thus, RNA-Seq indicated that MRTF significantly contributes to PKD-associated transcriptome alterations in our epithelial cell model system by enhancing cytoskeletal reorganization and matricellular proteins, which are key features of PEP. In addition, it regulates mitochondrial organization and metabolism and mitigates the acute innate immune response.

### 3.5. Integrin β1 (ITGB1) and MRTF-A Form a Feed-Forward Loop and Regulate Profibrotic Gene Expression

Our transcriptome sequencing data also indicated that PC loss stimulated the expression of ITGB1 (Figure 5A). Relevantly, the deletion of *ITGB1* dramatically reduced cystogenesis and fibrosis in *Pkd*1 fl/fl, Aqp2-Cre animals [66]. Further, *ITGB1* is a known direct MRTF target, whose promoter harbors a CARG box [67,68]. Using qPCR, we verified that PC1 and PC2 knockdown increased *ITGB1* mRNA expression. This transcriptional change was partially MRTF-A-dependent (Figure 5B). Moreover, ITGB1 *activity* was also affected by PC loss; using an activation-specific ITGB1 antibody (12G10), we observed a major increase in the number of active ITGB1 clusters, visualized as parallel thin lines in PC1- or PC2-silenced cells. A similar effect was observed upon the addition of manganese (Mn^2+^), a pan-integrin activator, which was used as a positive control (Figure 5C) [69]. Importantly, Mn^2+^ treatment alone was sufficient to prompt MRTF-A nuclear accumulation and robust *TAGLN* expression (Figure 5C–F). To assess the potential contribution of ITGB1 to MRTF translocation, we treated the cells with *ITGB1* siRNA, which resulted in a ≈70% drop in *ITGB1* mRNA (Figure 5E, left panel). This treatment efficiently suppressed the Mn^2+^-induced nuclear accumulation of MRTF (Figure 5E,F), and significantly but modestly mitigated PC1- or PC2-loss-provoked MRTF translocation (Figure 5G,H). Together, these findings imply that PC1/2 loss elevates the level and activity of ITGB1, and integrin activation is sufficient to induce MRTF translocation, which could contribute to (but is likely not indispensable for) this effect. Thus, while ITGB1 and MRTF can mutually activate each other, MRTF can also be triggered by other pathway(s) downstream of PC loss (see Discussion). 

### 3.6. The Role of MRTF in Paracrine Epithelial–Mesenchymal Communication upon PC Loss 

To assess whether PC loss can elicit a functional PEP state inducing fibroblast–MyoF transition [15,16,19,70]), and to test whether this might occur in an MRTF-dependent manner, we employed a bioassay. Conditioned media derived from control or siPC1-transfected epithelial cells were transferred onto fibroblasts, and the ensuing fibroblast–MyoF transition was determined based on the capacity of MyoF to contract and wrinkle the underlying soft substrate [54,71] (Figure 6A). The conditioned media of the siPC1-transfected cells induced a strong wrinkling capacity in the fibroblasts. Importantly, the conditioned media derived from cells exposed to the simultaneous knockdown of MRTF-A and PC1 lacked this effect (Figure 6B). Thus, PC loss induces the profibrotic paracrine PEP state in an MRTF-dependent manner.

### 3.7. The MRTF Pathway Is Activated In Vivo in Various Forms of PKD 

To assess whether PKD affects MRTF distribution in vivo, we analyzed two established adult-onset mouse models, *Pkd2* WS25/− and *Pkd1* RC/RC [44,55,56] (Figure 7A). *Pkd2* WS25/− is a compound heterozygous model, wherein the somatically unstable WS25 allele undergoes high rates of recombination, resulting in the loss of *Pkd2* and leading to early cystogenesis (within 3 months) [56]. *Pkd1* p.R3277C (RC) is a functional hypomorphic mutation, causing cystogenesis and fibrosis at 12 months of age. Because the younger (3 months) *Pkd2* WS25/− cohort showed more concordant results, this model was analyzed in more detail. The active RhoA levels were significantly higher in the *Pkd2* WS25/− kidneys compared to their corresponding controls (Figure 7B). As expected, the *Pkd2* WS25/− kidneys were histologically characterized by a large number of dilated tubular structures and cysts (Figure 7C, upper panes). Remarkably, robust nuclear MRTF accumulation was observed in a subset of epithelial cells, predominantly in dilated, precystic tubules (Figure 7C). Nuclear MRTF accumulation appeared to be much less and more homogenous in the control animals (Figure 7C). It is worth noting, however, that pathological tubular structures in the PKD animals were interspersed among normal ones (with low nuclear MRTF staining), reflecting the inherent variance in PKD1 dosage and *PKD2* genomic rearrangements, as suggested before [55,56]. In addition to tubular MRTF accumulation, there was a striking general (cytosolic and nuclear) upregulation of total cellular MRTF-A expression in the cyst-lining epithelium (Figure 7C and Appendix A). Concordant with these qualitative observations, both the total cortical (Figure 7D) and cystic (Figure 7E) MRTF expressions were significantly higher in the PKD2 kidneys than in the corresponding controls, and they were much higher in the cysts than in the normal regions of the PKD2 kidneys (Figure 7E). Similar observations (i.e., marked nuclear MRTF accumulation in the tubules and overexpression in the cysts) were made in the PKD1 animals as well (Figure 7C, lower panes, Figure 7D). In accordance with these results, the PKD patients’ samples also exhibited strong MRTF-A staining in the cystic epithelium (Appendix A). 

Considering that kidneys show focal heterogeneity in nuclear MRTF accumulation, we sought to compare this parameter quantitatively in the whole kidney sections of normal, PKD2, and PKD1 animals. Therefore, the distribution of nuclear MRTF-A (averaged for all animals tested with whole kidney slices, >300,000 cells/kidney) was measured by the HALO image analysis platform. A substantial shift towards higher nuclear intensities (OD) was observed in both PKD2 and in PKD1 kidneys, compared to the corresponding healthy controls (Figure 7F, Appendix A). The OD corresponding to the maximum of the distribution curve in the PKD mutant animals was selected as a threshold defining a ‘high’ MRTF-A nuclear presence. As shown in the insets, a significantly larger percentage of cells exhibited a ‘high’ nuclear MRTF-A expression in the PKD mutant kidneys than in the controls (Figure 7F). 

### 3.8. CTGF Expression Shows Spatial Correlation with Increased MRTF Expression 

Under physiological conditions, only stromal cells express CTGF in the kidneys. CTGF participates in ECM-associated profibrotic signaling by binding cell surface receptors (integrins), growth factors (TGFβ and BMPs), and ECM proteins (fibronectin) [72,73,74]. Further, both CTGF and PDGF expression are regulated by MRTF/SRF [29,32,75], constituting a positive profibrotic feedback loop in the epithelium. IHC quantification confirmed that CTGF and PDGF expression were elevated in the tubules of the PKD2 mutant animals compared to those of the control littermates (Figure 8A,B). 

We reasoned that, if MRTF directly drives *CTGF* transcription, the corresponding messages should show a spatial correlation; therefore, we compared *MRTF-A* and *CTGF* spatial mRNA expression on consecutive sections using RNAScope. *MRTF-A* mRNA expression was greatly upregulated in the dilated tubules, the adjacent stroma, and the cyst-lining epithelium of the PKD2 kidneys, contrasting the minimal expression in the normal tubules of the same animals and the control kidneys (Figure 8C–E).

Importantly, *MRTF*-positive epithelial cells ubiquitously expressed *CTGF* (Figure 8F–I), with a strong positive correlation between tubular *MRTF* and *CTGF* expression (Figure 8J). These tubules were typically localized in the vicinity of cysts and stromal areas with a strong *CTGF* expression. Altogether, the mRNA expression of *MRTF-A* and *CTGF* spatially overlapped, strengthening the potential role of MRTF in the epithelial initiation of profibrotic signaling.

## 4. Discussion

The current study reveals that the RhoA/cytoskeleton/MRTF pathway, a key initiator of epithelial fibrogenesis, is activated in PKD. Thus, MRTF emerges as a significant mediator of PKD-related fibrosis, one of the hallmarks of the disease. MRTF facilitates PEP and the consequent paracrine response, leading to fibroblast–myofibroblast transition in the adjacent stroma. From a broader perspective, PKD is a prominent example of epithelial-injury-initiated fibrogenesis, a process that has been shown to be sufficient for tubulointerstitial fiborosis [16,17,76,77,78,79].

Nuclear MRTF translocation is the direct consequence of PC-loss-induced RhoA activation and cytoskeleton remodeling. We showed that the knockdown of PC2, similarly to that of PC1 [44,76], activates RhoA. Increased RhoA activity can be due to the activation of RhoGEFs and/or the inhibition of RhoGAPs, and both mechanisms have been proposed in PKD [44,45]. How can PC1/PC2 loss affect these processes? One possibility involves heterotrimeric G proteins, particularly G_α12_, which is activated by PC1 loss [80,81,82,83,84], which is necessary for PKD pathogenesis [85] and can directly stimulate LARG, a GEF implicated in PC1-downregulation-induced RhoA activation [44]. A non-exclusive alternative entails ITGB1, which is overexpressed in PKD−/− animals, indispensable for cystogenesis and fibrosis [66], and can stimulate several GEFs (LARG and GEF-H1) [9,86,87,88]. We showed that the downregulation of PC1/2 activates ITGB1 and triggers its partially MRTF-dependent overexpression. 

While PC-loss-induced ITGB1 activation may not be a prerequisite for initial RhoA activation/MRTF induction, sustained ITGB1 activation/expression (promoted by increased cell contractility, stretching of the cyst wall, or ECM stiffening [9,89]) may signify a potent feed-forward mechanism. Furthermore, matricellular proteins, such as CTGF [90,91] and periostin [87,92,93], secreted by altered tubules and cysts, can directly activate other integrins, e.g., [94,95,96], and we showed that pan-integrin stimulation is sufficient to provoke strong MRTF nuclear translocation. Taken together, future studies should define the critical upstream RhoA-activating mechanisms in PKD and the contributions of various integrins in the initial and feed-forward activation of the RhoA/MRTF pathway. 

We show that dysregulated/lost PC signaling induces not only nuclear translocation, but also the overexpression of MRTF in tubular cells and the cyst-lining epithelium in vitro and in PKD1 and 2 mutant animal models. The underlying mechanisms are unknown, but it is worth mentioning that both β-catenin and YAP signaling are upregulated in PKD and other fibrotic diseases, and both can drive MRTF expression [91,97,98]. Conversely, MRTF is a direct inducer of TAZ expression [30,37]. Indeed, multilevel crosstalk exists between MRTF and YAP/TAZ, wherein these factors can target the same genes and regulate each other’s expression/activity/function [25,28,30,46,99,100], including MyF differentiation [97]. Thus, the MRTF-YAP/TAZ synergy constitutes yet another positive feedback loop. 

Our targeted gene expression studies and unbiased transcriptome analysis indicated that a substantial cohort of PC1/2 loss-regulated genes are MRTF-sensitive. These encode cytoskeletal components, matricellular proteins/fibrogenic mediators, and regulators of cell contractility and integrin signaling, many of which have been implicated in organ scarring. These findings are consistent with the previously demonstrated roles of MRTF (reviewed in [33]) and the prominence of SRF as a key transcriptional hub in PKD [50]. Moreover, our approaches assigned two novel functions to MRTF in this context. The first regards the expression of genes governing mitochondrial functions/metabolism. This new aspect requires further studies, given that PKD is characterized by robust metabolic changes (reduced OXPHOS and increased glycolysis) and alterations in mitochondrial shape and function [14,62,63,101,102]. The other possible function of MRTF is the *suppression* of inflammatory genes via the negative regulation of relevant transcription factors (e.g., NFκB, as raised before [60,61]). This may signify a role of MRTF as a switch between the inflammatory and the fibrotic aspects of PKD. In addition, the MRTF-dependent gene sets and GO categories, while overlapping, are not identical for PC1 vs. PC2 loss (Appendix A). This highlights the PKD-type specific roles of MRTF. While we started to unravel the spatial correlation between MRTF expression and fibrogenic cytokine production, spatial transcriptomics studies should extend these findings in the future. 

## 5. Conclusions

What is the significance of fibrosis in the overall pathology of PKD? A recent elegant report argues that fibrosis, per se, may inhibit cyst formation by mechanically restricting cyst growth, but it worsens survival [59]. These findings argue that combatting fibrosis is an important therapeutic option in PKD. Our proof-of-principle studies demonstrate that MRTF is activated upon PC loss and in PKD, and suggest that it acts as a significant mediator in the pathobiology of the disease. However, future functional studies should test how genetic or pharmacological interference with MRTF affects various aspects (inflammation, cytogenesis, and fibrosis) of the disease, and discern if MRTF is a viable drug target for the treatment of PKD. 

## Figures and Tables

**Figure 1 cells-13-00984-f001:**
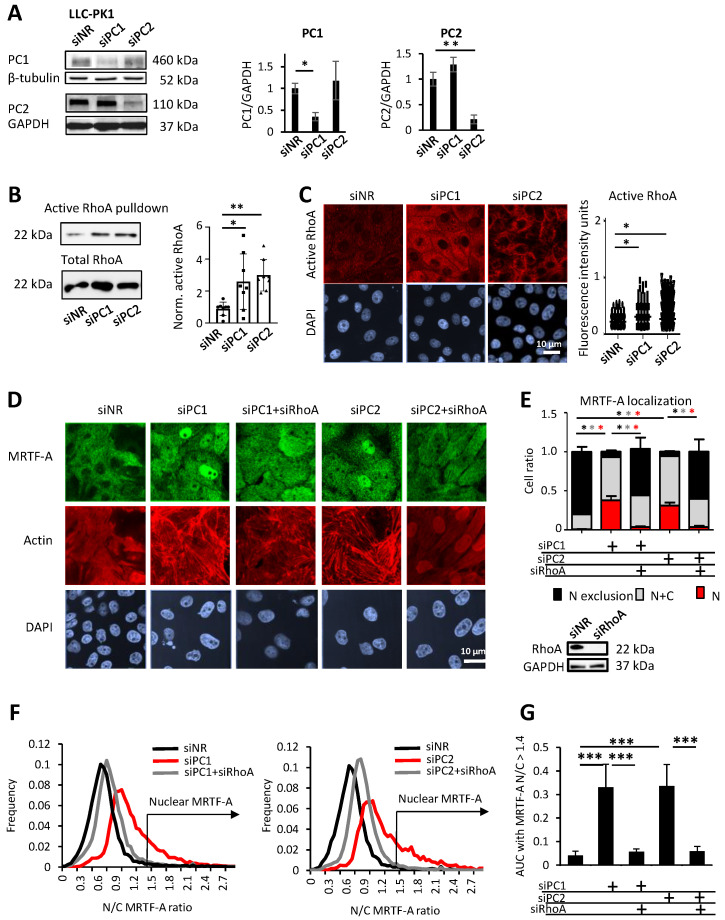
Loss of Polycystin 1 or 2 activates RhoA and leads to RhoA-dependent cytoskeletal remodeling and nuclear accumulation of MRTF-A. (**A**) PC1 or PC2 were silenced in LLC-PK1 cells with the corresponding siRNAs (150 nM and 100 nM, respectively, 48 h), without altering each other’s expression, as assessed by Western blot analysis. PC/GAPDH ratios are presented in the right panel. (**B**) Active RhoA was detected by GST-Rhotekin binding domain precipitation assay. Active RhoA precipitates and total cell lysate blots were probed with a RhoA-specific antibody. Fold change was normalized against the active RhoA/total RhoA ratio of control siRNA-transfected cells (non-related siRNA or siNR) (n = 8). (**C**) Cells were transfected as above. After 48 h, active RhoA was detected by immunofluorescence, using an antibody specific for the active, GTP-bound form. Representative images are shown for each indicated condition. Fluorescence intensity was quantified in individual cells using ImageJ. (**D**,**E**) PC1 (150 nM siRNA) or PC2 (100 nM siRNA) were silenced alone or concomitantly with RhoA (50 nM siRNA), as indicated. MRTF-A subcellular localization was assessed by immunofluorescent staining in each condition. In parallel, F-actin was visualized by staining with iFLUOR555-labelled phalloidin (representative images of the brightest plane are shown in **D**). Note that RhoA knockdown was highly efficient (**E**). Using visual assessment of MRTF-A subcellular localization, cells were grouped into the following three categories: 1. predominantly nuclear MRTF-A (red bars, N = nuclear), 2. even nuclear and cytoplasmic presence of MRTF-A (grey bars, N+C), and 3. nuclear exclusion of MRTF-A (black bars, N exclusion). * indicates *p* < 0.05, each colored * indicates the significance of the corresponding bar. (**F**) MRTF-A subcellular localization was analyzed by ImageXpress, a high-throughput automated digital imaging system. The distribution curves indicate the frequency of cells with increasing nuclear-to-cytoplasmic MRTF-A ratio (N/C). (**G**) Cumulative quantitation of the distribution curves shown in F, using N/C > 1.4 as a cutoff, which corresponds to definite nuclear MRTF-A localization by visual classification. * *p* < 0.05, **, *p* < 0.01, and *** *p* < 0.001.

**Figure 2 cells-13-00984-f002:**
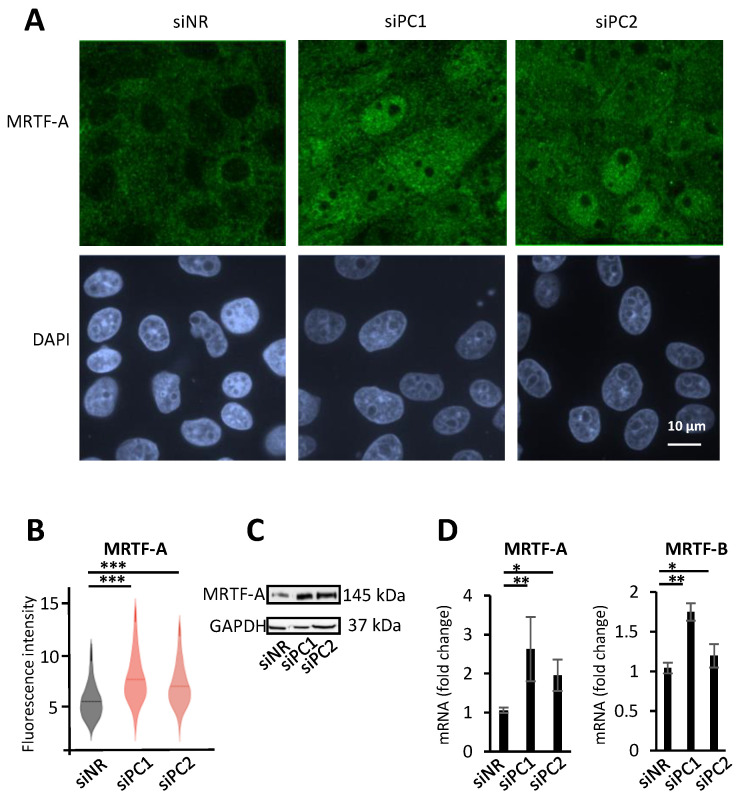
MRTF-A expression is upregulated by Polycystin loss. PC1 and PC2 were downregulated by siRNA transfection (150 nM and 100 nM, respectively, 48 h) and MRTF-A expression was assessed by three methods. (**A**,**B**) MRTF-A-specific fluorescence intensity was quantified by ImageXpress (see Section 2). Violin plots indicate the median fluorescence value. (**C**) MRTF-A protein expression was also visualized by Western blot analysis. (**D**) Both *MRTF-A* and *MRTF-B* mRNA expression was quantified by RT-qPCR. Results were normalized to *PPIA*. * *p* < 0.05, **, *p* < 0.01, and *** *p* < 0.001.

**Figure 3 cells-13-00984-f003:**
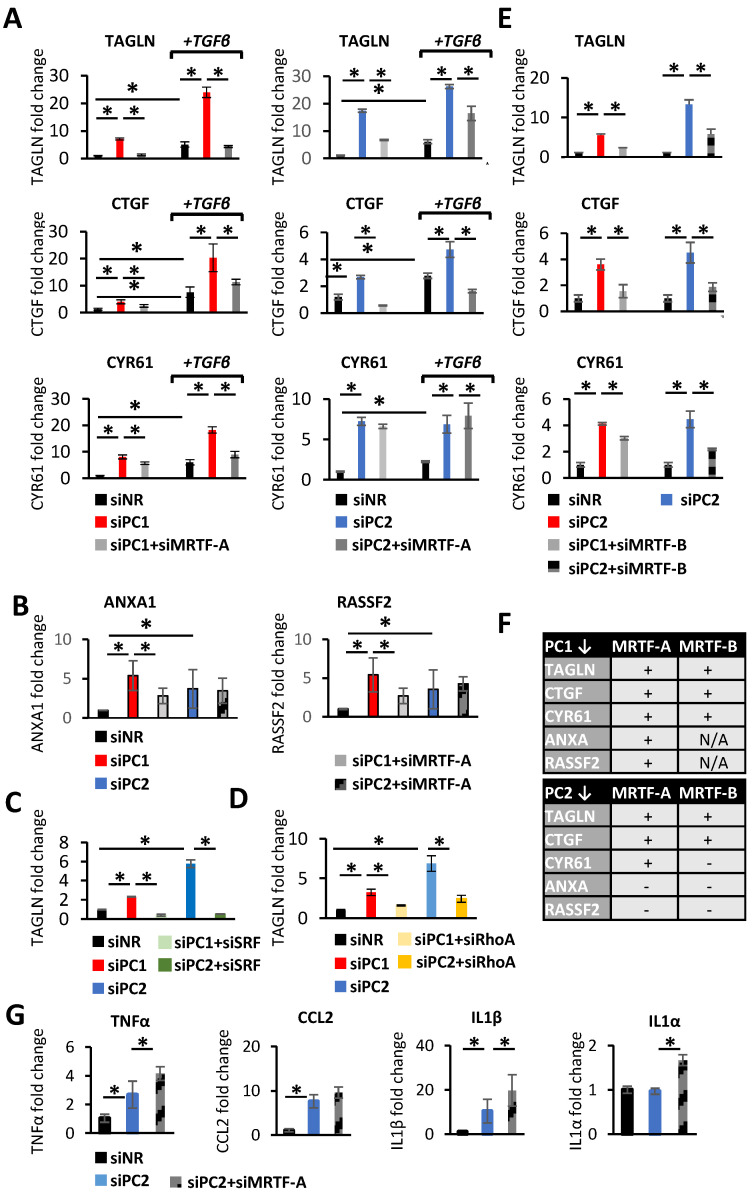
Polycystin loss induces the expression of profibrotic phenotype-related transcription program in a partially MRTF-dependent manner. (**A**–**E**,**G**) LLC-PK1 cells were transfected with control siRNA (NR) or siRNAs targeting *PC1* (150 nM), *PC2* (100 nM), *MRTF-A* (50 nM), *MRTF-B* (50 nM), *SRF* (100 nM), or *RhoA* (100 nM), as indicated. In (**A**,**B**), 24 h post-transfection, cells were treated with DMSO or 5 ng/mL TGFβ in serum-free media for an additional 24 h. Synergy/additional effects between the loss of PCs and the presence of TGFβ was assessed by measuring the expression of known PEP-related genes that are direct transcriptional targets of MRTF-A. The synergistic effects varied among genes, as shown. *ANXA* and *RASSF* expression was not stimulated by TGFβ treatment, but was stimulated by PC loss and was partially MRTF-driven. Fold changes (normalized to *PPIA* expression) were compared to the DMSO-treated, siNR-transfected samples (n > 4 in triplicates). (**C**,**D**) Knockdown of *RhoA* and *SRF* diminished the profibrotic effect of PC loss. (**F**) The table summarizes the MRTF-dependence of the investigated PEP-related and validated MRTF/SRF targets. (**G**) Silencing *MRTF-A* facilitates the PC loss-induced increase of some Nuclear factor κB (NFκB)-dependent inflammatory genes. mRNA abundance for tumor necrosis factor-α (*TNFα*), chemokine (C-C motif) ligand 2 (*CCL2*), and interleukin-1β and α (*IL****-****1β* and *IL-1α*) are shown. * *p* < 0.05.

**Figure 4 cells-13-00984-f004:**
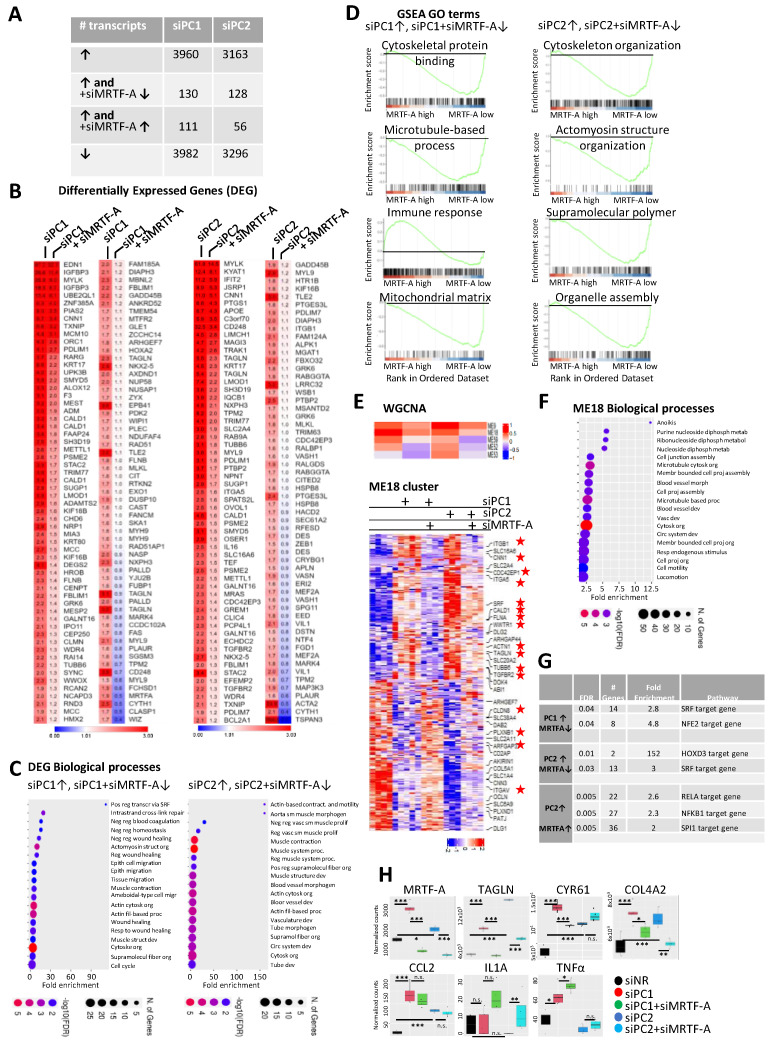
Next-generation transcriptome analysis of Polycystin- and MRTF-A-dependent gene expression. (**A**) Overview of experimental framework. LLCPK-1 cells were transfected with control siNRA (siNR, 200 nM, n = 3), siPC1 (150 nM, n = 4), siPC2 (100 nM, n = 4), alone, or with siMRTF-A (100 nM, siPC1+siMRTF-A n = 5, siPC2 n = 4). Gene expression was compared across the indicated conditions using RNA-Seq. 1.5-fold cutoff-defined upregulation and downregulation (log2 > 0.6 upregulated, log2 < −0.6 downregulated, *p* ≤ 0.05, base mean ≥50). The table indicates the number of transcripts upregulated upon PC knockdown (↑), and significantly changed when MRTF was also silenced. (**B**) DEG Analysis. Expression heatmap of transcripts that are upregulated by PC1 or PC2 loss (siPC1 and siPC2) in an MRTF-dependent manner (siPC1+siMRTF-A and siPC2+siMRTF-A). Numbers indicate fold changes compared to non-related siNR-transfected LLC-PK1 cells (siNR). (**C**) Shinygo 0.76 (a web application for R) was used to search for GO Biological Processes that are enriched when PC expression is silenced and are MRTF-dependent. (**D**) Gene Set Enrichment Analysis (GSEA) defined enriched GO terms in the above-described comparisons, as indicated. Selected significant GO terms are shown. Rank in Ordered Dataset is (Fold Change) × log_−10_ (*p*-value). (**E**) Expression heatmap of significant Weighted Gene Co-expression Analysis (WGCNA) modules that were MRTF-dependent and upregulated in PC1 or PC2 knockdown conditions. The heatmap of ME18 module is presented separately in the lower panel. Stars indicate genes that are known direct targets of MRTF. (**F**) The graph is the visual representation of significant GO biological processes related to ME18 module. (**G**) Predicted key transcription factors for genes that were upregulated upon PC knockdown and showed expression change upon MRTF-A silencing (Transfac TF Binding Site Enrichment Analysis). (**H**) Expression profiles from the RNA-Seq data shown for some individual genes under various conditions, as indicated. Key PKD-related genes (*MRTF-A*, *TAGLN*, *CYR61*, and *COL4A2*) and innate immunity-related genes (lower panels) are shown. Adjusted *p*-values are indicated: * *p* < 0.05, ** *p* < 0.01, and *** *p* < 0.001.

**Figure 5 cells-13-00984-f005:**
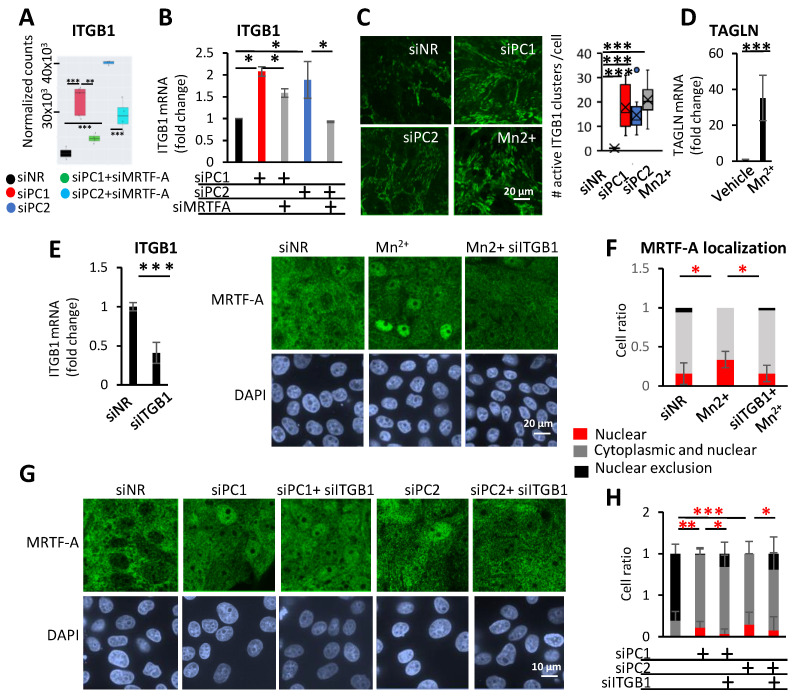
ITGB1 is overexpressed and activated upon Polycystin loss and regulates the subcellular localization of MRTF. (**A**,**B**) *ITGB1* mRNA expression was quantified using RNA-Seq and RT-qPCR in LLC-PK1 cells transfected with siPC (siPC1 150 nM or siPC2 100 nM) or siNR, and siMRTF-A (50 nM) or siNR, as indicated. RT-qPCR results were normalized to *PPIA* expression, against the siNR-transfected controls. (**C**) Left panel: Antibody, specific for the active conformation of ITGB1 (12G10), was used to visualize the clustering of active ITGB1. Right panel presents the number of ITGB1 clusters, normalized to cell number. (**D**) Mn^2+^ treatment (500 μM, 1 h) activated integrins and was sufficient to drive high *TAGLN* expression. Relative expression was measured by qRT-PCR and was normalized against *PPIA* expression. (**E**) *ITGB1* was partially silenced by specific siRNA (100 nM, 48 h). (**F**) Integrins were activated by Mn^2+^ treatment and MRTF-A subcellular localization was estimated by immunofluorescence staining. Using visual assessment (25 fields), MRTF-A localization was categorized as nuclear (red bars), even (grey bars), or cytoplasmic (black bars) (right panel). (**G**,**H**) Cells were transfected with siPC1 (150 nM) or siPC2 (100 nM) alone or together with siNR or siITGB1 (100 nM). Representative images of anti-MRTF-A immunofluorescent staining are shown (left panel) and reveal a modest but significant cytoplasmic shift upon ITGB1 silencing. MRTF-A localization was quantified as in (**D**, right panel). The images and quantification represent the results of a minimum of three independent experiments. * *p* < 0.05, **, *p* < 0.01, and *** *p* < 0.001.

**Figure 6 cells-13-00984-f006:**
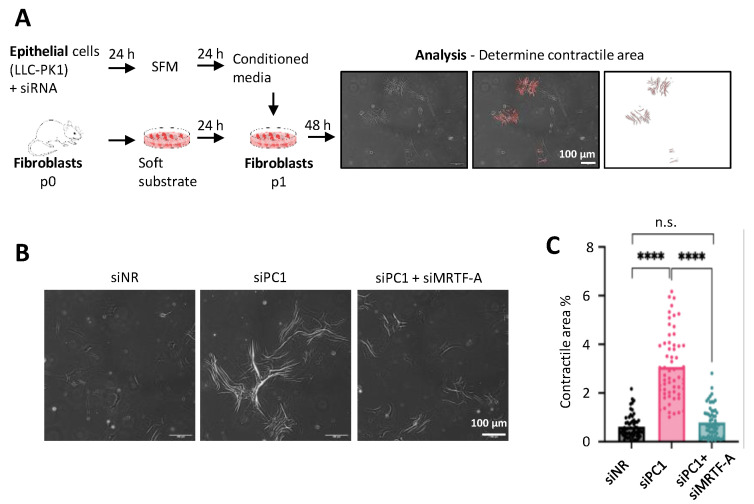
Epithelial Polycystin 1 loss induces MRTF-dependent paracrine signaling that potentiates fibroblast-to-myofibroblast transition. (**A**) Experimental design. LLC-PK1 cells were transfected with the indicated siRNAs (150 nM siPC1 and 50 nM siNR or siMRTF-A). Twenty-four hours post-transfection, cells were thoroughly washed to avoid the carry-over of siRNAs to fibroblast culture, and fresh serum-free media was added. Conditioned media was collected after an additional 24 h and was used to stimulate fibroblasts. Forty-eight hours later, >10 randomly selected areas of the fibroblast cultures were photographed under each condition. Cell contractile function, related to % area covered by wrinkles, was analyzed by FIJI ImageJ software and the values were normalized to cell number. Scale bars correspond to all images. (**B**,**C**) Representative images of fibroblasts cultures at the experimental end points and their quantification. Scale bars correspond to all images. **** *p* < 0.0001.

**Figure 7 cells-13-00984-f007:**
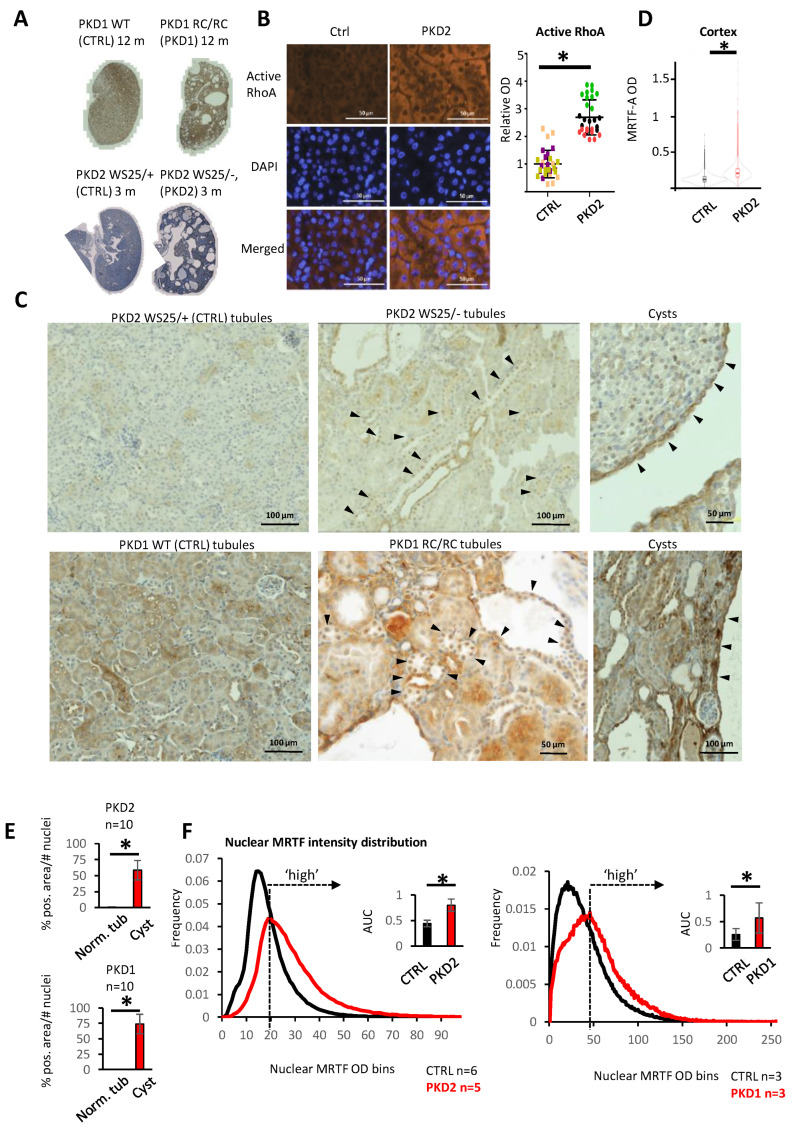
Transgenic PKD mouse model kidneys show MRTF nuclear accumulation and overexpression. (**A**) Twelve-month-old *Pkd1* RC/RC mice (PKD1, n = 3), WT control littermates (Ctrl, n = 3), 3-month-old *Pkd2* WS25/− animals (PKD2, n = 6), and *Pkd2* WS25/+ control littermates (Ctrl, n = 6) were evaluated. At the selected time points, micro- and macrocysts were present in both models, except for one of the *Pkd2* WS25/− mice. This animal was omitted from the analysis. (**B**) RhoA activation was assessed by detecting the GTP-bound form by immunofluorescence. Ten fields were quantified from each animal (n = 3, right panel). (**C**) Nuclear MRTF expression was quantified using HALO Area Quantification module. Arrowheads point at cells with nuclear MRTF-A expression (middle panels) or strong MRTF-A overexpression in the cystic wall (right panels). Scale bar corresponds to all images. (**D**) IHC assessed MRTF subcellular localization in both PKD mouse models and control animals. Arrowheads indicate nuclear MRTF-A in the tubules and increased MRTF-A expression in the cystic epithelial wall. (**E**) Whole kidney sections were analyzed, and MRTF-A/DAB average OD was individually reported for the >300,000 nuclei per animal, using the inbuilt Object Quantification module of HALO. OD values were assigned to 256 incremental bins, and single-nuclei-associated OD data are presented as distribution curves. The highest-frequency OD bin in the PKD animals was assigned as the threshold for ‘high’ nuclear MRTF-A accumulation (dotted arrows). The inserts depict the AUC ratio corresponding to nuclei with ‘high’ nuclear MRTF-A. (**F**) MRTF-A expressions on histologically ‘normal’ tubules and cysts were compared within the PKD2 WS25/− and PKD1 RC/RC kidneys. Automatic Threshold Method (ATM score 0 or 1, inbuilt Area Quantification module of HALO) was used to dichotomize DAB-positive and -negative pixels. MRTF-A positive area was normalized to the number of nuclei in each examined tubule and cyst. * *p* < 0.05.

**Figure 8 cells-13-00984-f008:**
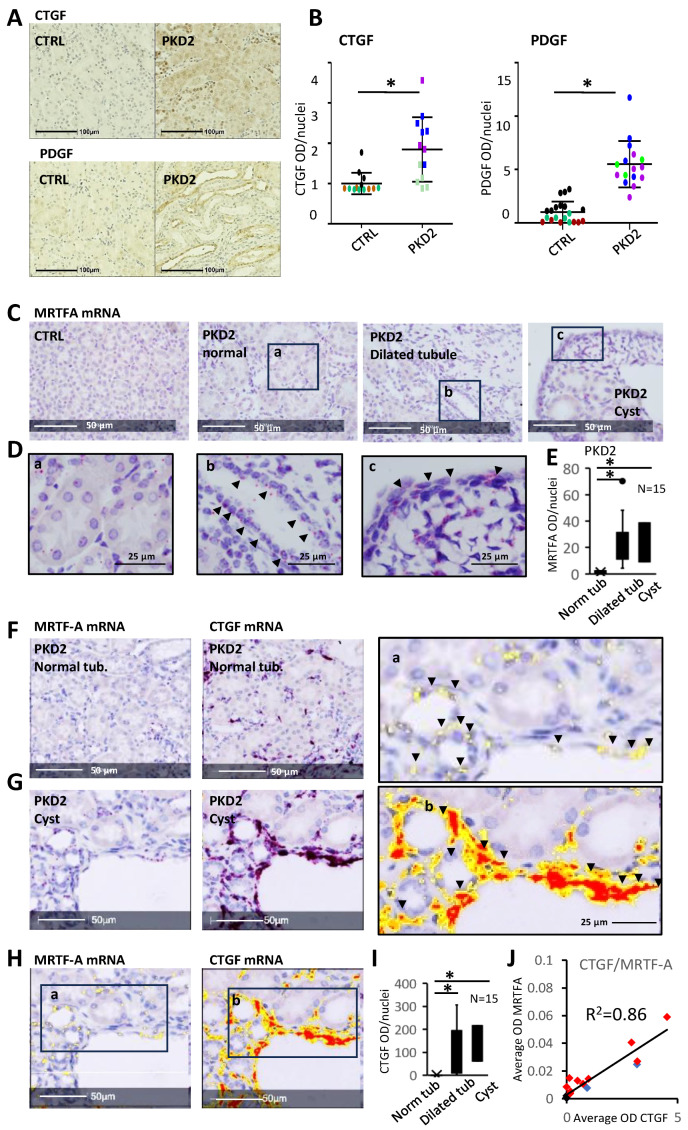
Fibrogenic cytokine expression is increased in PKD, and *CTGF* expression spatially correlates with *MRTF* mRNA abundance in vivo. (**A**,**B**) CTGF and PDGFB expression was detected in *Pkd2* WS25/− (PKD2) and *Pkd2* WS25/+ animals (Control, Ctrl) by IHC (n = 3). Expression was quantified in 4–10 fields per animal using the Area Quantification package of HALO, normalizing for the number of nuclei. (**C**–**I**) RNAScope was used to detect *MRTF-A* and *CTGF* mRNA expression in *Pkd2* WS25/− and *Pkd2* WS25/+ animals. Images in (**D**) are enlarged areas of normal and dilated tubules and a cyst. Arrowheads point to *MRTF* expression signals. HALO Area Quantification module was trained to differentiate the red signal on the hematoxylin-stained background. For easy visualization, positive signals are shown as yellow, orange, and red colors, corresponding to a low, moderate, and high signal intensity (H and enlarged areas). RNAscope signal intensity was measured by the Area Quantification module and was normalized to the number of nuclei (**E**,**I**). (**J**) Correlation of *CTGF* and *MRTF* expression is shown in normal and dilated tubules and cysts of *Pkd2* WS25/− animals. * *p* < 0.05.

**Table 1 cells-13-00984-t001:** siRNA and qPCR primer sequences.

	**siRNA Sequences**
siPC1	GCG CUG ACA GAG UUG GAC AUA (dT) (dT)
siPC2	GAC CGU GAG AGA UAC CUU AAA (dT) (dT)
siRhoA	AGC AGG UAG AGU UGG CUU U (dT) (dT)
siMRTF-A	CCA AGG AGC UGA AGC CAA A (dT) (dT)
siMRTFB	CGA CAA ACA CCG UAG CAA A (dT) (dT)
siITGB1	CUG AAG AAG UAG AGA UAA U (dT) (dT)
siSRF	GGA ACU GUG CUG AAG AGU A (dT) (dT)
siPC1 4331	UGU CAA GCC GCG UGA AUA A (dT) (dT)
siPC2 1644	CAA GAU UGA UGC AGU GAU A
	**qPCR Primer Sequences**
Porcine CTGF F	GTG AAG ACA TAC CGG GCT AAG
Porcine CTGF R	GAC ACT TGA ACT CCA CAG GAA
Porcine GAPDH F	GCA AAG TGG ACA TGG TCG CCA TCA
Porcine GAPDH R	AGC TTC CCA TTC TCA GCC TTG ACT
Porcine PPIA F	CGG GTC CTG GCA TCT TGT
Porcine PPIA R	TGG CAG TGC AAA TGA AAA ACT G
Porcine MRTF-A-E7F	TAT CCT GCC TGT GGA GTC CA
Porcine MRTF-A-E8R	ATA AGG CGT CAC TGC TGT CC
Porcine MRTFB-E6F	GCA GGC CAC TCA GAT GAA GT
Porcine MRTFB-E7R	GTC ACT GCT GTC CTC GTC AA
Porcine IFNB1-F	ATG AGC TAT GAT GTG CTT CGA TAC
Porcine IFNB1-R	GAA TTG TGG TGG TTG CAT AAT C
Porcine TNFA-F E2-3	CCA ATG GCA GAG TGG GTA TG
Porcine TNFA-R E2-3	CTG AAG AGG ACC TGG GAG TAG
Porcine IL1A-F E8-9	GCA ACT TCC TGT GAC TCT AAG
Porcine IL1A-R E8-9	GCG GCT GAT TTG AAG TAG TC
Porcine CCL2-F	CAG CAG CAA GTG TCC TAA AG
Porcine CCL2-R	TCC AGG TGG CTT ATG GAG TC
Porcine TAGLN E3 F	GAG CAG GTG GCT CAG TTC TT
Porcine TAGLN E4 R	CCA CGG TAG TGT CCA TCA TTC
Porcine CYR61 E3F	AAA GGC GGC TCC CTG AAG
Porcine CYR61 E4 R	GAC GTG GTT TGA ACG ATG C
Porcine RASSF1 E2 F	GAG TGG GAG ACA CCT GAC CT
Porcine RASSF1 E3 R	ATA CAG GAC GCA CCA GCT TC
Porcine ANXA 1 E5 F	TTG CTG AAA ACT CCA GCT CA
Porcine ANXA 1 E7 R	GCA AAG CCT TCT GAT AAT CTC C

## Data Availability

RNA transcriptome data are submitted to the GEO repository under #GSE252716.

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
