# Peer review of "Myocardin-Related Transcription Factor Mediates Epithelial Fibrogenesis in Polycystic Kidney Disease"

_cells, 2024, doi:10.3390/cells13110984_

Round 1

Reviewer 1 Report

Comments and Suggestions for Authors

This study explores how the loss of Polycystin 1 or 2 (PC1/2) in Polycystic Kidney Disease (PKD) activates the RhoA/cytoskeleton/MRTF pathway, leading to fibrosis. The pathway involves RhoA activation, cytoskeletal changes, and increased MRTF expression. The authors conducted extensive experiments with well-defined experimental design.

1. However, in Figure 1C and D, there are co-stainings with RhoA. In such cases, pseudo-color should be applied to the fluorescent images instead of using mono-color staining to enhance clarity and interpretation.

2. Applying pseudo-color in Figure 2A would benefit readers and improve understanding of the image.

3. Figure 4 does not meet publishable quality standards due to unreadable legends and data. The authors should replace this with a higher resolution image for clarity and readability.

4. Figure 5 C, E, and G are the same.

5. Figure 6 B is the same.

6. In Figure 7 D (box plot), there is a statistically significant difference, but there is considerable data out of box plot. Consequently, it is challenging to discern a statistically significant difference between the control and PKD2 groups.

7. In Figure 8J, it should display the R-squared (R²) value, not just "R".

Author Response

Thank you very much for the positive review and the suggestions.

Here are our point-by-point answers. 

1 and 2: We used pseudo-color in the figures, as advised.

3 and 5:  We increased the resolution of the figures (low resolution was due to a suboptimal PDF conversion) and applied pseudo-color, as suggested. 

6. The data are presented as a violin plot for better visibility. 

7. The R value has been replaced by R square.

Thank you for your comments again. 

Reviewer 2 Report

Comments and Suggestions for Authors

This is a through investigation on the mechanisms of fibrogenesis in PKD. It is well designed and well presented.

The main question of this research was whether MRTF (RhoA/cytoskeleton/myocardin-related transcription factor (MRTF) is activated by Polycstin 1 or 2 loss /loss of function and whether it plays a critical role in fibrogenic reprogramming of the epithelium pathway as an emerging mediator of epithelium-induced fibrogenesis.   This is of interest since polycystic kidney disease is characterized by extensive cyst formation and progressive fibrosis.

The specific gap in the field is that the molecular mechanisms weherby the loss/los-of function of polycystin 1 or 2 provokes fibrosis is largely unknown.
The data presented add to the knowledge of the mechanisms involved in fibrosis in polycystic kidney disease

I do not think that improvements of the design, the methods used or the description of the data are necessary.

The conclusions are concise and correspond to the presented data.
The references are appropriate.

Author Response

Thank you very much for your evaluation and opinion.